# SATPose: Improving Monocular 3D Pose Estimation with Spatial-aware Ground Tactility

## ABSTRACT

Estimating 3D human poses from monocular images is an important research area with many practical applications. However, the depth ambiguity of 2D solutions limits their accuracy in actions where occlusion exits or where slight centroid shifts can result in significant 3D pose variations. In this paper, we introduce a novel multimodal approach to mitigate the depth ambiguity inherent in monocular solutions by integrating spatial-aware pressure information. To achieve this, we first establish a data collection system with a pressure mat and a monocular camera, and construct a large-scale multimodal human activity dataset comprising over 600,000 frames of motion data. Utilizing this dataset, we propose a pressure image reconstruction network to extract pressure priors from monocular images. Subsequently, we introduce a Transformer-based multimodal pose estimation network to combine pressure priors with monocular images, achieving a world mean per joint position error (W-MPJPE) of 51.6mm, outperforming state-of-the-art methods. Extensive experiments demonstrate the effectiveness of our multimodal 3D human pose estimation method across various actions and joints, highlighting the significance of spatial-aware pressure in improving the accuracy of monocular 3D pose estimation methods. Our dataset is available at: https://anonymous.4open.science/r/SATPose-51DD.

## CCS CONCEPTS

• **Human-centered computing → Ubiquitous and mobile computing systems and tools**; • **Computing methodologies → Motion capture**.

## KEYWORDS

Multimodal 3D Human Pose Estimation, Pressure Sensor, Multimodal Human Activity Dataset, Pressure Image Reconstruction

## 1 INTRODUCTION

Estimating 3D human poses from monocular images and videos [21, 42, 48, 53, 62] is a classic and challenging computer vision task, which involves determining the locations of various body keypoints to construct a representation of the human skeleton. Thanks to its wide range of applications in sports and fitness, medical rehabilitation, virtual character embodiment in games, etc., significant efforts have been made to explore advanced network architectures

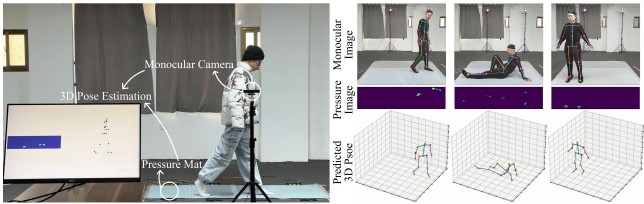

**Figure 1: We constructed a real-time 3D human pose estimation system, consisting of a monocular camera and an optional pressure mat (left). With pressure images, we mitigate the inherent depth ambiguity in monocular vision solutions, enabling accurate prediction of 3D human poses (right).**

[20, 29, 43, 56] to improve its accuracy. However, the fact that monocular vision cannot reliably gauge depth makes it difficult for these efforts to be fully effective [26, 56]. Specifically, different body movements can be projected into the image space in the same 2D pose. During certain actions like sideways poses, some parts of the human body may be occluded by other body parts; slight shifts in the body centroid that may not be easily discernible from monocular images can lead to different body poses; deficiencies in depth information estimation can also detrimentally impact the effectiveness of global displacement estimation. To alleviate the impact of depth ambiguity, previous research either resorts to the temporal context captured by successive action frames [26, 39] or the spatial context of the input monocular image modeled by Graph Neural Networks (GNNs) [8, 20]. Although effective, these methods rely solely on monocular 2D visual cues and lack true 3D spatial awareness, thus limiting their performance.

Parallel to monocular-vision-based methods, grounding tactility has also been used to estimate 3D human poses. The benefit of this approach originates from the large size and flexible nature of pressure sensors, enabling their embedding into everyday exercise equipment such as fitness mats. This expands the applicability of pressure sensors in various domains including sports, fitness and medical rehabilitation, etc. To achieve these, most existing works rely on array-type pressure sensors as tactile sensing devices, which are highly sensitive to external forces and offer precise localization of force areas and perception of force magnitude. For example, some prior works [6, 45] utilized pressure insoles and pressure mats to study the distribution of plantar pressure during human motion; Luo et al. [32] proposed a method utilizing solely pressure mat information to estimate 3D human pose. Despite achieving lower full-body estimation accuracy compared to visual approaches, pressure sensors demonstrated excellent performance in lower limb joint estimation, highlighting their potential to address the depth ambiguity inherent in vision-based solutions. Inspired by this, we propose an integrative approach that ingeniously leverages the sensitivity of pressure sensors to grounding positions to mitigate the

depth ambiguity of monocular vision method, thereby improving the accuracy of 3D human pose estimation. However, this is a challenging task as: i) to the best of our knowledge, there are no existing systems that can simultaneously collect both tactile and vision data, while collecting this type of multimodal dataset requires substantial effort. ii) Developing an algorithm to fuse the different modalities introduced by the new system poses another challenge. iii) Introducing new hardware increases the system's complexity.

To tackle these challenges, in this paper, we established a human activity collection system incorporating a pressure mat and a monocular camera, and introduced a Transformer-based multimodal framework designed for accurate 3D human pose estimation from both monocular images and pressure images (Fig. 1). Compared to monocular methods, our approach enhances the 3D perception of the system by integrating additional spatial knowledge from ground pressure, resulting in more accurate 3D pose estimation. Specifically, utilizing the data collection system we built, we first created a large-scale multimodal human activity dataset, namely the PVM (Pressure, Vision, Mocap) Dataset, which includes pressure images, monocular images and ground truth 3D poses. This dataset covers data from 20 volunteers performing 16 different actions, totaling over 600,000 frames. Particularly, to enhance the generalizability and performance of our method in various scenarios, especially those where a pressure mat is unavailable, we proposed a test-time adaptation strategy that reconstructs pressure images from monocular images and feeds them into the pose estimation network. Leveraging the large-scale PVM Dataset, we can reliably reconstruct ground pressure images from monocular images, and then achieve accurate 3D human pose estimation even if no real pressure information is available during testing. Experimental results show that our multimodal 3D pose estimation network achieved a world mean per joint position error (W-MPJPE) of 51.6mm and 51.8mm with real pressure and predicted pressure, respectively, both surpassing state-of-the-art methods. This demonstrates the effectiveness and flexibility of our approach, allowing for choosing whether to use the pressure mat depending on the requirement for higher accuracy or simpler system complexity.

In summary, our contributions can be summarized as follows:

- A novel human activity collection system including a pressure mat and a monocular camera. Leveraging this system and optical motion capture cameras, we constructed the PVM Dataset, a large-scale multimodal human activity dataset.
- A test-time adaptation strategy that predicts pressure data from monocular images, enabling accurate 3D human pose estimation solely from monocular images during testing. This makes our system applicable to more diverse real-world scenarios, especially those without a pressure mat.
- A Transformer-based multimodal fusion framework that incorporates spatial-aware pressure information to mitigate the inherent depth ambiguity of monocular-vision-based 3D pose estimation methods and improve their accuracy.

## 2 RELATED WORK

### 2.1 Vision-based 3D Human Pose Estimation

3D human pose estimation (HPE) is a traditional computer vision task [1, 2, 38, 48, 62], existing works of 3D HPE can broadly be classified into two mainstreams. The first involves the one-stage estimation [12, 38, 46], where the 3D poses are directly predicted from the images. The second is the two-stage estimation, where the 2D poses are first extracted from the images, and then a lift from 2D to 3D is performed. As for two-stage estimation, benefiting from the excellent performance of state-of-the-art 2D pose detectors [7, 19, 34], numerous works [25, 29, 39, 44, 49, 59, 61] engage in improving the performance of 2D-3D pose lifting. Most of them can be divided into TCN (Temporal Convolutional Network)-based methods [29, 39], GCN (Graph Convolutional Network)-based methods [8, 20, 54–56, 60], and Transformer-based methods [18, 26–28, 31, 41, 43, 49, 63]. Compared to TCN-and-GCN-based structures, Transformer-based architecture is better suited for modeling long sequences of 2D poses due to its well-designed attention mechanism. In this paper, we innovatively incorporate a multimodal Transformer-based structure to fuse the 2D poses extracted from monocular images and ground pressure data.

### 2.2 Application of Pressure Sensors

Pressure sensors can provide comprehensive biomechanical information. Their integration in various fields such as interactive control [14, 36, 40], robot touch [24, 57], gesture recognition [3, 15, 30, 47, 51, 52], etc., highlights their unique advantages. For example, TouchEditor [58] supports text editing for AR glasses through tactile gestures on a flexible touchpad. SmartSleeve [36] enables real-time sensing of surface and deformation gestures on a textile pressure sensor. In addition to the aforementioned small-area pressure sensors, studies by Clever et al. [9] and Casas et al. [5] utilize 2D pressure data to simulate lying postures, while Luo et al. [32] propose a method for 3D pose estimation using pressure sensors integrated into a mat, showing promising prospects in depth estimation and lower limb pose estimation. However, it is difficult to provide effective support for upper limb posture estimation using a single pressure sensor. Therefore, we integrate visual and tactile information, leveraging the complementary nature of both to enhance the pressure-based pose estimation, while addressing issues such as depth information absence in monocular images and estimation errors from centroid shifts.

### 2.3 Multimodal 3D Human Pose Estimation

Recent research [10, 17, 23, 33, 37, 64] indicates that integrating data from various sensors can improve the accuracy of 3D human pose estimation. For instance, Pan et al. [35] merge monocular RGB and sparse IMU data to facilitate robust motion capture even when visual signals are unavailable. [16] proposes an architecture fusing RGB and LiDAR data for precise pedestrian localization and pose prediction. [37] achieves efficient and accurate pose tracking by integrating data from LiDAR and IMU sensors. Additionally, Zhou et al. [65] integrate visual features, skeletal poses, probability maps, and multi-channel audio signals to create a hybrid representation for human action analysis. However, these wearable device-based solutions introduce foreignness and interference to human movement in rehabilitation and sports training. Sensor displacement during movement can also compromise the effectiveness of pose estimation. In contrast, our solution can offer users a seamless training experience while ensuring stable tracking performance.

## 3 PVM DATASET

In this section, we detail the experimental setup employed for the collection of the large-scale multimodal human activity dataset, PVM (Pressure, Vision, Mocap) Dataset.

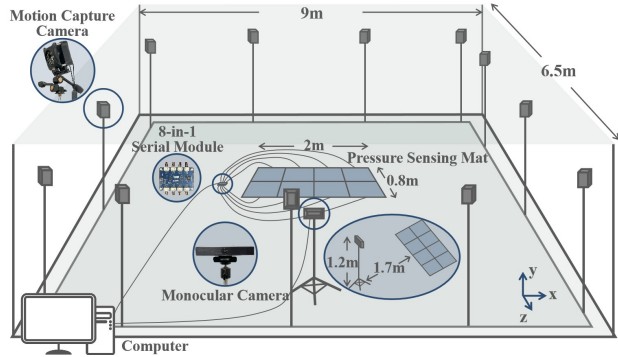

**Figure 2: Diagram of the multimodal data collection system, comprising a pressure mat and a monocular camera situated in an optical motion capture space, and all devices centrally controlled by a computer.**

### 3.1 Participants and Actions

*3.1.1 Participants.* Twenty volunteers (10 M and 10 F, mean age = 22.9, SD = 2.86) were invited to participate in our data collection experiment. The body mass index (BMI) of participants ranged from 18 to 28, ensuring a moderate amount of body shape variability and different ranges of mobility among them. Participants were outfitted in a motion capture suit crafted from elastic Velcro fabric, including a top, pants, hat, and foot covers. The suit was designed to be flexible and elastic, prioritizing participants comfort and minimizing interference with their movements to the greatest extent possible. Our research received Institutional Review Board (IRB) approval from the local institution of the university. All individuals signed an informed consent form before the experiment, and they were remunerated accordingly after the completion of the experiment.

*3.1.2 Actions.* We focus on 3D human pose estimation for actions with some body parts being occluded and actions sensitive to slight shifts in body centroid. It is challenging to address these issues with monocular images due to depth ambiguity, yet they can impair the effectiveness of applications requiring high-precision pose estimation, such as sports training. Thus, by filtering and integrating existing human activity datasets [22, 32] and the functional movement screen (FMS) action set [11], we designed a set of 16 actions (see Fig.3). Participants performed each action for approximately 90 seconds (1800 frames), with the flexibility to take breaks as needed. Prior to the commencement of the experiment, the researcher conducted a brief demonstration of each action for the participant. Once the experiment began, no additional guidance was provided, allowing participants the freedom to perform actions at any location according to their individual preferences.

### 3.2 Apparatus

*3.2.1 Pressure Sensing Mat.* The ground pressure images were collected using a commercial, large-format, high-resolution pressure

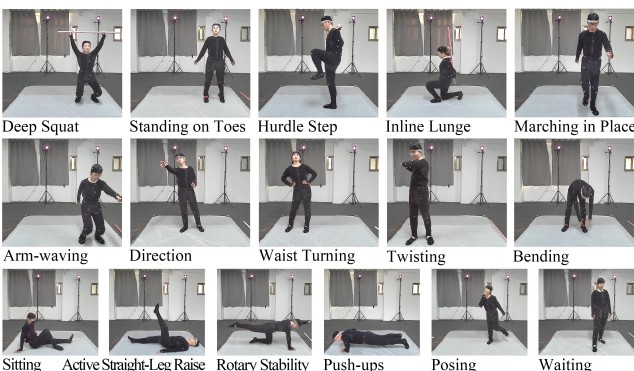

Deep Squat · Standing on Toes · Hurdle Step · Inline Lunge · Marching in Place

Arm-waving · Direction · Waist Turning · Twisting · Bending

Sitting · Active Straight-Leg Raise · Rotary Stability · Push-ups · Posing · Waiting

**Figure 3: 16 human actions in the PVM dataset.**

sensing mat provide by *Matrix Innovation*[1]. As shown in Fig.4, this mat is equipped with eight independent piezoresistive film sensors. The application of external pressure leads to a slight change in the distance between film layers, resulting in variations in the resistance and output voltage of the sensing layer. The pressure sensors can detect pressures of up to 30 kPa, with a sensitivity of 0.5 kPa. Each pressure sensor covers an area of $50 \times 40 \text{cm}^2$ with a resolution of $64 \times 32$. Thus, the final assembled pressure sensing mat has an area of $200 \times 80 \text{cm}^2$, and the resolution of output pressure image is $256 \times 64$. The output values for each sensor point fall within the range of [0, 100]. These eight pressure sensors are wired and connected to an eight-in-one serial module, which is further connected to a computer. Sensor data is transmitted via serial communication at a collection frame rate of 20Hz, and data synchronization among sensors is achieved through broadcast communication.

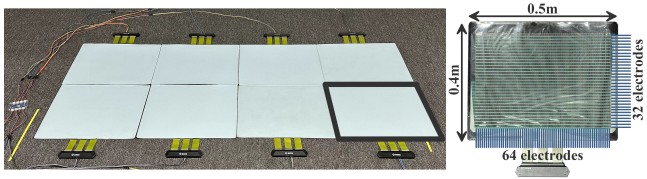

**Figure 4: The pressure mat is composed of eight piezoresistive film sensors, with each pressure sensor having an area of 0.5 $\times$ 0.4m$^2$ and a resolution of 64 $\times$ 32.**

*3.2.2 Monocular Camera.* We employed the C93 monocular camera manufactured by *AONI*[2] to record videos of human activities (Fig.2). The videos were captured at a resolution of 1280 $\times$ 720 with a frame rate of 30Hz. The camera was securely mounted on a tripod, positioned 1.7m directly in front of the pressure sensing mat at a height of 1.2m. This camera setup was designed to ensure the capture of the complete body of all participants at any location on the pressure sensing mat. The camera was connected to a computer through a wired connection, facilitating control over video recording and the transmission and storage of the video stream.

---

[1]https://www.moxiantech.com/
[2]https://www.aoni.cc/

*3.2.3 Optical Motion Capture System.* We utilized a optical motion capture (Mocap) system manufactured by *NOKOV*[3], which consists of 12 Mars2H series optical cameras, to record the 3D human pose (Fig.2). This data serves as the ground truth for subsequent model training. The Mocap system determines the 3D positions of the 18 body joints by capturing the reflective markers on human body, ultimately recording the human motion with millimeter-level precision (error range of ±0.15 mm) and a frame rate of 60Hz.

## 3.3 Dataset Structure

The three aforementioned data acquisition devices were connected to the same computer for data collection and storage control, ensuring data synchronization. The computer receives eight $64 \times 32$ pressure matrices from the pressure mat and ultimately concatenates them into a $256 \times 64$ pressure matrix. Mocap data is exported in .bvh format and further converted into the 3D world coordinates of various body keypoints, resulting in data of size $18 \times 3$ for each frame. For monocular videos, we retain the original image frames, and extract 2D skeleton keypoints with the open-source human pose estimation library, OpenPose [4]; Finally, we obtain 2D skeleton with 18 keypoints corresponding one-to-one with the 3D one. After post-processing, data from all modalities were unified to a frame rate of 20Hz. In the end, we collected over 600,000 frames of multimodal human activity data from 20 participants performing 16 actions. The dataset was divided into a training set and a test set with a ratio of 16:4 participants.

## 4 METHOD

In this section, we introduce our data-driven pressure image reconstruction method along with the multimodal 3D human pose estimation framework.

## 4.1 Data-driven Pressure Image Reconstruction

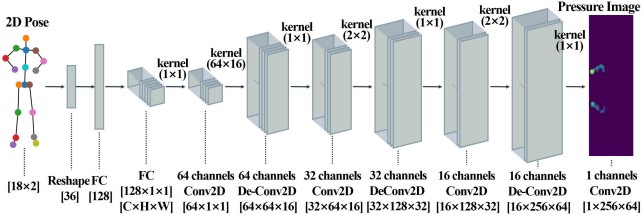

**Figure 5: Architecture of the pressure image reconstruction network, employing a convolutional neural network to predict the pressure image from input 2D human pose.**

*4.1.1 Motivation.* The spatial-aware grounding tactility is crucial for enhancing the 3D perception capability of monocular-vision-based human pose estimation networks. However, the reliance on the pressure mat may limit the application of this system to more scenarios. There is a need for an adaptive method that can utilize pressure information during testing while overcoming the restrictions associated with the pressure mat. A dual-pronged strategy involves extracting pressure priors from 2D poses, and subsequently utilizing both 2D poses and pressure images reconstructed from

³https://www.nokov.com/



**Figure 6: Qualitative evaluation results of the pressure image reconstruction model.**

2D poses to estimate 3D poses. The PVM dataset, a large-scale multimodal human activity dataset including rich pressure images corresponding to 2D poses, provides a robust data foundation for training a high-precision pressure image reconstruction network.

*4.1.2 Implementation.* We propose a pressure image reconstruction network based on a multi-layer deconvolution neural network (Fig.5). Given a 2D pose $\tilde{Po} \in \mathbb{R}^{J \times 2}$, representing the (x, y) coordinates of $J(J = 18)$ body joints, we first concatenate these $J$ 2D coordinates to obtain $\tilde{Po}' \in \mathbb{R}^{J \cdot 2}$. Then, through a Multi-Layer Perception (MLP), we embed $\tilde{Po}'$ into a high-dimensional feature $\tilde{Z} \in \mathbb{R}^{dm}(m = 128)$, and further map it to the image space $\tilde{Z}' \in \mathbb{R}^{dm \times 1 \times 1}$ using a fully connected layer. Subsequently, the feature $\tilde{Z}'$ is input into multiple deconvolution blocks, gradually reducing its channel number, upsampling its resolution, and finally obtaining the reconstructed pressure image $I \in \mathbb{R}^{1 \times H \times W}$. $H = 256$ represents the height and $W = 64$ represents the width of the image. Each deconvolution block comprises a convolutional layer for channel transformation, a deconvolutional layer for upsampling image resolution, three convolutional layers for further feature extraction, and corresponding batch normalization, dropout, and ReLU activation operations. Additionally, Mean Squared Error (MSE) is used as the loss function for model optimization. In this process, the network learns how to synthesize more detailed pose features, ultimately successfully reconstructing pressure images of the target size. This achievement is attributed to the large-scale PVM dataset, enabling the data-driven network to effectively extract pressure priors and accurately reconstruct pressure images.

*4.1.3 Training.* We train the pressure image reconstruction model for 35 epochs with one 3090 GPU, taking about 12 hours to converge. We employ the Adam optimizer with a peak learning rate of 1e-3 that gradually decreases following a cosine learning rate schedule.

*4.1.4 Reconstruction Performance.* We assessed the performance of the pressure image reconstruction model on the PVM test set. Fig.6 presents visual results illustrating the qualitative evaluation of the reconstruction model. The model demonstrated outstanding reconstruction performance in various scenarios, including actions with both feet on the ground, actions with a single foot on the ground, and actions involving multiple body parts in contact with the ground. Notably, the model excelled in actions with a frontal stance and both feet on the ground, accurately reconstructing the position and shape of footprints, and capturing differences in pressure distribution caused by changes in the body centroid (*Deep Squat* and *Waiting*). For lateral and single-footed actions, reconstructed pressure images also accurately reflected the body centroid

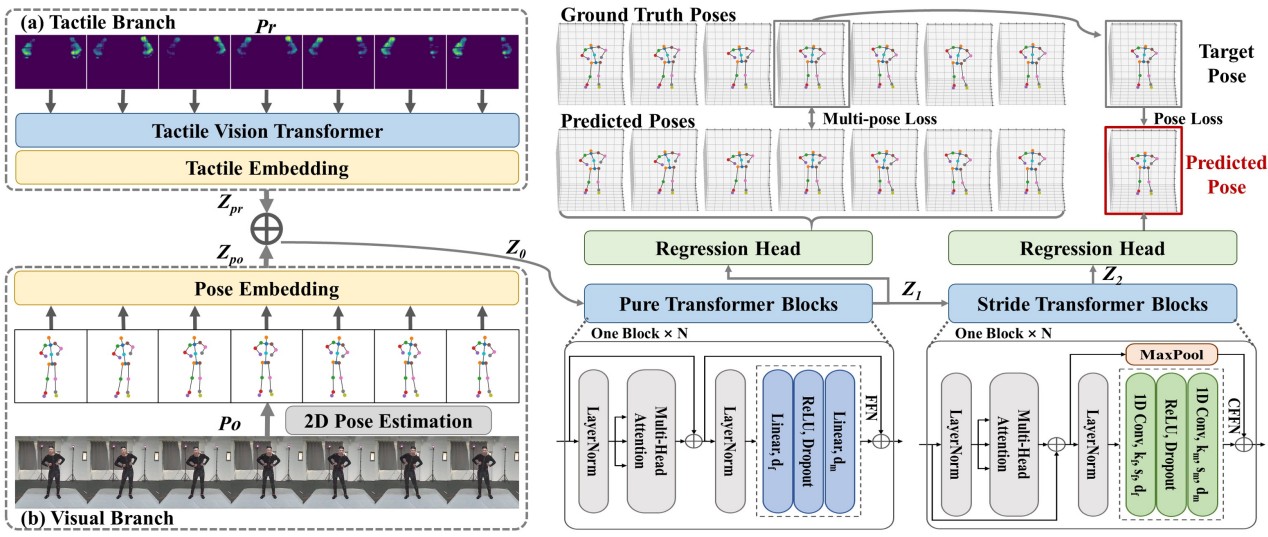

**Figure 7: Architecture of the 3D human pose estimation network, employing a Transformer-based neural network to predict the 3D human pose of the target frame from input sequences of pressure images and monocular images.**

and provided a reasonably accurate depiction of the contact position (*Inline Lunge* and *Hurdle Step*). Even for actions involving multiple body parts in contact with the ground, such as *Push-ups*, the model achieved satisfactory reconstruction results. In more flexible actions with greater individual variations, such as *Sitting*, the model appropriately reconstructed the pressure distribution. In summary, the reconstructed pressure images possess the capability to provide the necessary spatial and pressure magnitude information for 3D pose estimation.

## 4.2 Multimodal 3D Human Pose Estimation

We propose a Transformer-based framework (Fig.7) for estimating 3D pose from sequences of 2D keypoints and pressure images.

*4.2.1 Pose Embedding.* The encoding of 2D poses is executed with a convolutional network. The input consists of multiple frames of 2D poses, denoted as $Po \in \mathbb{R}^{N \times J \times 2}$, where $N = 351$ represents the number of frames, and $J = 18$ represents the number of body joints. The $N$ frames encompass the current frame along with the past $\frac{N-1}{2}$ frames and future $\frac{N-1}{2}$ frames. The pose embedding concatenates the 2D coordinates $(x, y)$ of the $J$ joints for each frame, resulting in $N$ tokens $Po' \in \mathbb{R}^{N \times (J \cdot 2)}$. Subsequently, a 1D convolutional layer embeds each token into a high-dimensional feature $Z_{po} \in \mathbb{R}^{N \times d_m}$, $(m = 256)$.

*4.2.2 Pressure Embedding.* The encoding of pressure images is realized through a Vision Transformer (ViT)-based architecture [13]. The input pressure images, denoted as $Pr \in \mathbb{R}^{N \times H \times W}$, have $N$ frames in the sequence, with $H = 256$ representing the height and $W = 64$ representing the width of each pressure image. Prior to the encoding, pressure images are resized from $256 \times 64$ to $120 \times 120$, aligning them with the input format of Transformer. Subsequently, each pressure image is divided into a sequence of tokens, which are then fed into Transformer's encoder. ViT encodes this information into a high-dimensional feature representation,

resulting in $Z_{pr} \in \mathbb{R}^{N \times d_m}$. Leveraging the global relationship modeling capability of ViT for sequential data, it effectively captures spatial relationships within pressure images.

*4.2.3 Pure Transformer.* The 2D pose embedding and pressure image embedding are added to obtain a fused feature vector, denoted as $Z_0 \in \mathbb{R}^{N \times d_m} = Z_{po} + Z_{pr}$. $Z_0$ is further encoded using Vanilla Transformer [50] to obtain an intermediate feature vector $Z_1 \in \mathbb{R}^{N \times d_m}$. Vanilla Transformer operates based on self-attention mechanisms, allowing it to better understand the spatial relationships between different joints and contextual information between different frames. By stacking these layers, the network can effectively encode information about multi-frame human movements, capturing the spatiotemporal correlations between body keypoints.

*4.2.4 Stride Transformer.* Our ultimate goal is to estimate the 3D human pose for the current frame. Therefore, we use Stride Transformer [26] to compress the frame dimension of the features $Z_1$. Unlike Pure Transformer, Stride Transformer replaces the feedforward neural network (FFN) layer with a convolutional feedforward neural network (CFFN) layer. This involves downsampling the input vector through convolutional operations, resulting in the final feature representation $Z_2 \in \mathbb{R}^{1 \times d_m}$ for the current frame.

*4.2.5 Regression Head and Loss Function.* To better model multi-frame pose information, we supervise the model at both the full sequence scale and the single target frame scale. Specifically, regression is performed on the outputs $Z_1$ and $Z_2$ of Pure Transformer and Stride Transformer, respectively, to obtain the estimated 3D poses $X_1 \in \mathbb{R}^{N \times J \times 3}$ and $X_2 \in \mathbb{R}^{J \times 3}$. $X_1$ and $X_2$ represent the estimation results for the 3D poses across $N$ frames and the 3D pose for the current frame, respectively. The regression head consists of a batch normalization layer and a 1D convolutional layer. Given the 3D pose sequence ground truth $Y_1 \in \mathbb{R}^{N \times J \times 3}$ and the target frame 3D pose ground truth $Y_2 \in \mathbb{R}^{J \times 3}$, the L2 norm is computed between $X_1$ and $Y_1$, and between $X_2$ and $Y_2$, serving as two

**Table 1: Quantitative comparison results with state-of-the-art methods on protocol #0. Bold and underline indicate the best and second-best values.**

| Protocol #0 | D.S. | S.T. | H.S. | I.L. | M.P. | A.W. | Direction | W.T. | Twisting | Bending | Sitting | A.S.L.R. | R.S. | P.U. | Posing | Waiting | **Avg.** |
|---|---|---|---|---|---|---|---|---|---|---|---|---|---|---|---|---|---|
| Shan et al. [44] MM2021 | 56.7 | 47.1 | 57.5 | 60.7 | 55.0 | 60.5 | 52.4 | 52.6 | 51.3 | 57.7 | 52.4 | 54.6 | 87.8 | 76.2 | 70.1 | 62.3 | 59.7 |
| Li et al. [26] ToMM2022 | 49.8 | 43.1 | 58.5 | 65.8 | 54.5 | 56.2 | 50.9 | 46.9 | 55.5 | 60.4 | 52.2 | 50.9 | 73.9 | 70.4 | 65.1 | 49.7 | 56.5 |
| Shan et al. [43] ECCV2022 | 48.7 | 46.4 | **55.7** | 60.0 | 49.3 | 55.2 | 51.5 | 44.4 | 50.7 | 57.7 | 50.4 | 50.9 | 71.2 | 68.1 | 62.8 | 70.1 | 55.8 |
| Li et al. [27] CVPR2022 | 52.8 | 47.2 | 58.2 | 61.6 | 55.5 | 56.1 | 57.3 | 47.3 | 58.1 | 59.4 | 55.5 | 53.8 | 76.3 | 67.4 | 63.8 | 72.2 | 58.9 |
| Zhang et al. [59] CVPR2022 | 64.0 | 62.1 | 60.0 | 64.1 | 66.0 | 59.6 | 64.0 | 63.9 | 55.5 | 55.4 | 50.7 | 54.0 | 72.5 | **57.7** | 80.1 | 69.6 | 62.5 |
| Zhao et al. [61] CVPR2023 | 53.6 | 48.5 | 56.6 | 64.1 | 58.8 | 61.2 | 58.1 | 47.9 | 52.1 | 64.6 | 51.6 | 53.6 | 81.3 | 65.9 | 74.1 | 67.3 | 60.0 |
| Yu et al. [56] ICCV2023 | 50.3 | 44.7 | 56.2 | 64.4 | 52.7 | 54.2 | 51.5 | 50.2 | 58.1 | 57.5 | 52.3 | 56.1 | 79.2 | 70.0 | 60.4 | 48.9 | 56.7 |
| Ours (pred. pressure) | 47.1 | 41.4 | 57.0 | 63.0 | 48.1 | **48.4** | 46.2 | 43.1 | 45.4 | 54.7 | 49.8 | 47.7 | 68.5 | 62.9 | 56.4 | **45.4** | 51.8 |
| Ours | **44.4** | **41.2** | 56.5 | 61.6 | 47.4 | 53.8 | 46.9 | 40.5 | 44.2 | 56.4 | 47.2 | 46.1 | 70.7 | 63.6 | 58.8 | 46.9 | **51.6** |

separate loss functions, *i.e.*, $L_1(Y_1, X_1) = \sum_{n=1}^{N} \sum_{j=1}^{J} \left\| Y_{1j}^n - X_{1j}^n \right\|_2$, $L_2(Y_2, X_2) = \sum_{j=1}^{J} \left\| Y_{2j} - X_{2j} \right\|_2$. Therefore, the total loss for the 3D pose estimation network is $L = L_1 + L_2$.

*4.2.6 Training.* We train the pose estimation model for 25 epochs with one 3090 GPU, which takes about 20 hours to converge. We employ the Adam optimizer with a peak learning rate of 1e-3 that gradually decreases following a cosine learning rate schedule.

## 4.3 Metrics

MPJPE (mean per joint position error) is a commonly used metric for evaluating the performance of human pose estimation networks. In this paper, the variant of MPJPE is used, *i.e.*, **W-MPJPE (protocol #0)**, measured in millimeters (mm). It indicates the Euclidean distance between the predicted joint positions and the true joint positions in the world coordinate system.

## 5 EXPERIMENTS

## 5.1 Comparison with State-of-the-Art Results

We compared our multimodal method for 3D human pose estimation with state-of-the-art (SOTA) monocular-vision-based methods. Utilizing the official code of these methods, we performed training and testing on our PVM dataset. Table 1 presents the results of the 16 actions for these SOTA methods and our method on protocol #0. The results of our method encompass estimations based on real pressure and predicted pressure.

Our estimations based on real pressure (51.6mm) and predicted pressure (51.8mm) both outperform SOTA methods, achieving optimal results in terms of the mean values and for the majority of actions. Despite the challenges in some complex actions, such as "Rotation Stability", where various methods exhibit poor performance (around 80mm), our approach consistently delivers a relatively favorable result (below 70mm). This underscores the superiority of our multimodal pose estimation method. Additionally, the test results based on predicted pressure demonstrate performance comparable to those based on real pressure. This presents a reliable approach for obtaining prior pressure information during testing, mitigating the constraints of pressure mats and allowing adaptation to a broader range of scenarios.

Fig.8 illustrates qualitative comparison results for four actions. The causes of estimation errors include: i) Information loss due to body parts being obscured (*e.g.* the knee joints during bending); ii)

Depth ambiguity resulting in errors in estimating joint positions in depth (*e.g.* the feet joints during bending and waiting), while our method has achieved significant improvement in the z-axis direction (depth direction) of 4.7mm compared with the monocular method; iii) Subtle variations in the body centroid that are challenging to capture from 2D vision (*e.g.* the shoulder-neck joints during bending and the hip joints during waist turning), while our method has achieved significant improvement ranging from 5 to 10mm in these joints compared with the monocular method; iv) Occasional deviations in the horizontal direction due to global displacement, such as the feet joints during waist turning.

In summary, the spatial perception characteristics of pressure images offer significant gains in the depth information required for 3D human pose estimation through the following mechanisms: i) Facilitating the localization of various body joints in contact with the ground and providing accurate global displacement information by utilizing the position distribution of pressure; ii) Reflecting the body centroid through the differences in pressure magnitude across these distributions, aiding in inferring the spatial relationships between different body joints.

## 5.2 Accuracy across Joints

Fig.9 depicts box plots illustrating estimation errors for various joints across different action distributions, showcasing errors along the x, y, and z axes for each joint. The multimodal human pose estimation network was adjusted by separately removing pressure embedding and 2D pose embedding, and models were trained with only 2D poses and only pressure inputs on the PVM dataset.

Methods excel in estimating lower limb joints compared to upper limb joints. This is attributed to the more complex joint structure and increased degrees of freedom in the upper body. Additionally, for upper limb joints, estimation errors gradually increase from the root joint (hip joint) to the middle joints (shoulder, elbow joints), and then to the end joints (wrist joints). Similarly, the end joints exhibit a larger error range, showing significant performance variations under different actions. This complexity stems from the motion transfer through the multi-joint chain, involving rotations, translations, or their combinations, making end joints more flexible in terms of degrees of freedom.

Methods based on real pressure (see Fig.9a) and predicted pressure (see Fig.9b) exhibit comparable performance, both outperforming results of the monocular-vision-based method in Fig.9c. This is primarily evident in the following aspects: i) Significant reduction

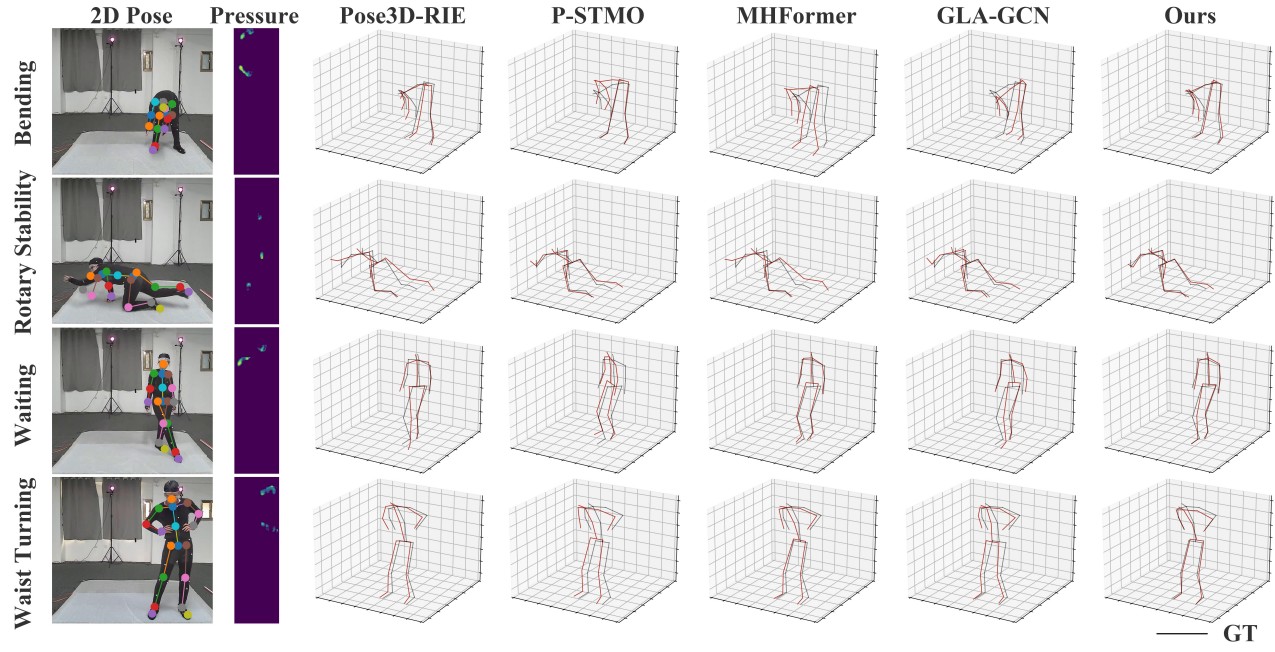

**Figure 8: Qualitative comparison results with state-of-the-art methods, Pose3D-RIE [44], P-STMO [43], MHFormer [27], GLA-GCN [56]. Skeletons drawn with black line represent the ground truth poses.**

in estimation errors across all joints, notably with a 5.9mm decrease in the upper limb joints and a 4.2mm decrease in the lower limb joints; ii) Error reduction is more noticeable in joints closer to the root node, with a 6.0mm decrease in the hip, followed by sequential decreases from the knees to the toes (3.4mm, 3.3mm, and 3.1mm) and from the shoulders to the wrists (6.0mm, 3.4mm, and 2.5mm); iii) The z-axis direction, representing the depth dimension, demonstrates the most substantial error reduction (4.7mm) in predictions compared to the other two axes (1.7mm and 2.3mm).

On one hand, ground pressure sensors capture pose-related information by sensing the contact between the human body and the ground. As a result, they complements reliable and stable data for pose estimation of the feet and leg joints. When it comes to upper limb joints, it plays a significant role in estimating the positional relationships between joints, thereby substantially enhancing the accuracy of upper limb joints. On the other hand, introducing spatial-aware pressure knowledge helps mitigate the inherent limitation of depth information loss, resulting in notable reduction in estimation errors along the z-axis. In addition, the purely pressure-based method demonstrates the weakest performance overall (120.9mm) but exhibits relatively superior and stable performance for lower limb joints (83.6mm) compared to upper limb joints (158.3mm) (see Fig.9d). This showcases its potential for estimating lower limb joints.

## 5.3 Ablation Studies

*5.3.1 Different Modal Combinations.* In Table 2, we have summarized the 3D human pose estimation errors for different modal combinations. The method that integrates 2D poses with reconstructed pressure and the one that integrates 2D poses with real pressure

achieved comparable results, surpassing the performance of the purely monocular-vision-based approach. This outcome strongly validates the efficacy of introducing spatial-aware pressure information in enhancing the performance of monocular-vision-based 3D human pose estimation methods.

**Table 2: Pose estimation errors across modal combinations.**

| Modal Combinations | Pressure | 2D Pose | 2D Pose & Pred. Pressure | 2D Pose & Pressure |
|---|---|---|---|---|
| **Protocol #0 (mm)** | 120.9 | 56.7 | 51.8 | 51.6 |

*5.3.2 Different Pressure Image Resolutions.* We conducted tests on pressure images with varying resolutions. Specifically, we trained four models, each utilizing a different size when resizing the input pressure images originally at a resolution of $256 \times 64$. The four sizes employed for resizing were $40 \times 40$, $80 \times 80$, $120 \times 120$, and $160 \times 160$. The results in Table 3 indicate that as the resolution of the pressure images increases, there is a general trend of decreasing error in 3D human pose estimation. Higher-resolution pressure images provide more detailed information, enabling the model to capture the nuances of human pose more accurately. However, it is noteworthy that when the image resolution increases from $120 \times 120$ to $160 \times 160$, the estimation error does not decrease but rather increases. Beyond a certain resolution level, additional details in the images may not be crucial for the pose estimation task and could instead lead to difficulties in network convergence. Moreover, higher-resolution images may escalate computational costs and model complexity. Therefore, we ultimately adopted the $120 \times 120$ resolution scheme for optimal performance.

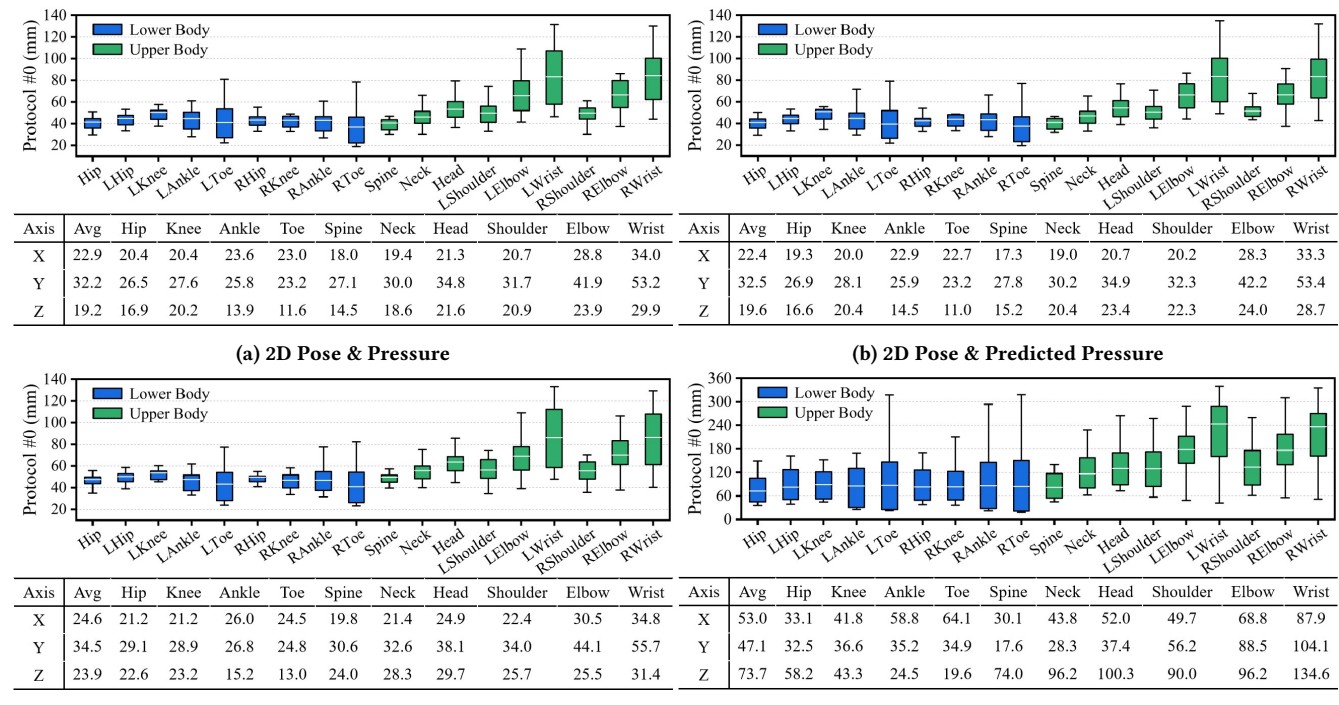

**Figure 9: Estimation errors for each joint (calculated using Euclidean distance) and estimation errors along the x, y, and z axes for each joint (calculated using Manhattan distance) for different modal combinations, all computed as global distances.**

**Table 3: Pose estimation errors across pressure resolutions.**

| Pressure Resolutions | 40*40 | 80*80 | 120*120 | 160*160 |
|---|---|---|---|---|
| Protocol #0 (mm) | 54.0 | 53.1 | 51.6 | 52.7 |

*5.3.3 Different Dataset Sizes.* We also conducted ablation studies on different dataset sizes. To mitigate the impact of individual and action variations on the results, we retained data for 16 actions from 20 participants in the sub-datasets. Random sampling was performed at different proportions for each action of each individual in the PVM dataset, namely 0.1, 0.25, 0.5, 0.75, and 1.0. As depicted in Table 4, the pose estimation network trained on a larger-scale dataset achieved superior results. This outcome underscores the significance of a large dataset for accurate 3D human pose estimation. A large dataset can provide ample samples, diverse pose variations, and detailed information, enabling the model to learn and generalize more effectively.

**Table 4: Pose estimation errors across data sizes.**

| Data Sizes | 0.1 | 0.25 | 0.5 | 0.75 | 1 |
|---|---|---|---|---|---|
| Protocol #0 (mm) | 61.2 | 58.6 | 54.9 | 52.5 | 51.6 |

## 6 CONCLUSION

In this paper, we propose a novel 3D human pose estimation method that integrates ground pressure images with monocular images. As a foundation, we construct a large-scale multimodal dataset incorporating pressure images, monocular images, and ground truth 3D poses recorded during 16 distinct actions from 20 participants, totaling over 600,000 frames. Furthermore, to accommodate situations where a pressure mat is unavailable, we propose a pressure image reconstruction network capable of reconstructing ground pressure images from monocular images. Experimental results indicate that both combining monocular vision with real pressure and combining monocular vision with predicted pressure outperformed all monocular-vision-based SOTA methods. This demonstrates the exceptional performance of pressure information in enhancing the accuracy of 3D human pose estimation task, and provides viable new directions for future research in this field.

**Limitations and Future Work:** For pressure embedding, we conduct image-wide feature extraction, which, although effective, entails redundant information. Exploring more compact feature extraction methods may enhance information density, potentially improving estimation accuracy and efficiency for the multimodal pose estimation network. For pressure image reconstruction, there are various methods to introduce pressure priors. In this paper, we employ a data-driven image reconstruction approach, yet future exploration could involve biomechanical analysis to understand force relationships and generate pressure images, offering an intriguing and promising direction for future research.

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
