# OpenReview forum: "SATPose: Improving Monocular 3D Pose Estimation with Spatial-aware Ground Tactility"
_acmmm.org/ACMMM/2024/Conference — MM2024 Poster_

### Official Review · Reviewer_er6z · 2024-05-17

**Rating:** 4
**Confidence:** 3

**Summary:**

This paper proposes a  method to conduct 3D human pose estimation with a pressure mat and a monocular camera. Their designed system can be used in sports in local space. Furthermore, thy construct a new dataset for evaluating their model. Obvious enhancement can be found in their experiments.

**Strengths:**

1. Dataset contribution should be encouraged.
2. The system can be used in some real-world applications.
3. This paper is easy to understand.
4. The distance ambiguity in 3D pose estimation is a challenging problem.
5. The experiments show that their model can improve the performance.

**Limitations:**

1. To verify the efficience of each component, more ablation experiments should be added.
2. In the real-world applications, the position of pressure mat maynot be put in the place as the dataset. There maybe some offsets from the place in the dataset. Can this problem affect the model performance?
3. I think that author should  also report MPJPE, which is applied more commonly.
4. It seems the dataset is not so challenging, due to the limited number of person, angle of view, and so on.
5. If the test-time adaption can be extend to multi-person scene, this work will be more valueable.

**Suitability:**

2

---

### Official Review · Reviewer_6bDt · 2024-05-23

**Rating:** 4
**Confidence:** 4

**Summary:**

This paper introduces a novel approach that integrates ground pressure images with monocular images to enhance the accuracy of 3D human pose estimation. The authors construct a large-scale multimodal dataset and propose a pressure image reconstruction network for scenarios where a pressure mat is unavailable. The experimental results demonstrate good performance over SOTA approaches.

**Strengths:**

Multimodal Dataset Construction: The creation of a dataset that includes ground pressure images, monocular images, and ground truth 3D poses enhances the generalizability of the model.
Pressure Image Reconstruction Network: This network addresses the limitation of requiring pressure mats, making the approach more flexible and widely applicable.
Performance Metrics: The method has been evaluated against existing SOTA methods, showing good improvements in 3D pose estimation accuracy.

**Limitations:**

-	Duplicated Information in feature Embedding: The image-wide feature extraction method for pressure embedding can lead to redundant information, which may affect efficiency and accuracy.
-	Simpler Pressure Reconstruction: The data-driven approach for pressure image reconstruction lacks biomechanical insights, which could further refine the accuracy by understanding force relationships.
-	Limited Evaluation Metrics: While the paper shows improvements in error metrics, it could benefit from additional evaluation metrics to provide a more comprehensive assessment of the model's performance.
-	Upper Limb Performance: Despite improvements, the method still shows relatively weaker performance for upper limb joints compared to lower limb joints.
-	Resolution Sensitivity: The accuracy of pose estimation varies with different pressure resolutions, indicating that selecting the optimal resolution is critical and application-specific.
-	Real-Time Applicability: The computational complexity of combining monocular vision with pressure data might affect real-time application, requiring further optimization for practical usages.

**Suitability:**

3

---

### Official Review · Reviewer_4bnQ · 2024-05-24

**Rating:** 4
**Confidence:** 4

**Summary:**

Estimating 3D human poses from monocular images is challenging due to depth ambiguity, especially in cases of occlusion or slight centroid shifts. This paper introduces a multimodal approach that integrates spatial-aware pressure information to improve accuracy. The authors created a large dataset using a pressure mat and a monocular camera, and developed a pressure image reconstruction network to extract pressure priors. A Transformer-based multimodal pose estimation network combines these priors with monocular images, achieving stoa performance. The proposed method demonstrates the importance of spatial-aware pressure in enhancing monocular 3D pose estimation.

**Strengths:**

- The technical contribution of this paper is significant, especially the collection of a new dataset containing the pressure modality.
- The authors also provide a large number of experiments and ablation studies to demonstrate the performance of proposed method.
- The paper writing is clear and easy to follow.

**Limitations:**

- The network framework is a straightforward integration of previous methods (e.g., ViT, stride transformers). This is not a major limitation, as a significant portion of the technical contribution stems from the dataset.

- This paper evaluates only the W-MPJPE metric, without assessing the MPJPE metric (with joint root aligned) or the PA-MPJPE metric (MPJPE with Procrustes analysis).

- The evaluation methodology for the W-MPJPE metric is unclear. Specifically, is the world coordinate the same as the camera coordinate in a monocular motion capture setup? Furthermore, what is the coordinate system when using pressure information and images as simultaneous inputs?

- There is a concern about whether the method relies on the shape of the pressure mattress and if it can generalize well to different shapes of pressure mattresses.

- It is also uncertain if the pressure prediction network performs well with subtle human movements, such as when the feet are slightly above the ground. If it does not, will this negatively impact the final predicted pose when only the RGB image is used as input?

**Suitability:**

3

---

### Official Review · Reviewer_zJ8S · 2024-05-25

**Rating:** 5
**Confidence:** 4

**Summary:**

The paper presents a multimodal approach that integrates both vision and pressure data to address depth ambiguity in monocular 3D human pose estimation. To achieve this, the authors developed a data collection system that combines a pressure mat and a monocular camera, paired with marker-based motion capture ground truth, resulting in a comprehensive multimodal human activity dataset. The proposed system highlights the significance of spatial-aware pressure information in enhancing the accuracy of monocular 3D pose estimation methods.

**Strengths:**

1. Interesting Idea: The integration of pressure information with vision data to jointly alleviate depth ambiguity is innovative and demonstrates significant improvements over existing methods.
2. Beneficial Dataset: The proposed PVM dataset seems highly beneficial to the relevant community, and provides a robust foundation for training and evaluating the proposed models.
3. Non-trivial effort went into conducting extensive experimental validation.
4. The paper is well-structured and easy to follow.

**Limitations:**

1. Limited Effectiveness for Upper Body: The proposed paradigm seems to alleviate depth ambiguity primarily for the lower body, with limited usefulness for the upper body.
2. Lack of Evaluation for Test-Time Adaptation: The authors proposed a test-time adaptation strategy that predicts pressure data from monocular images, which is important for adapting to existing monocular settings. However, the manuscript lacks evaluation of this module within existing state-of-the-art methods, which would demonstrate its ability to enhance these methods. Without this evaluation, the novelty of the proposed approach is somewhat diminished.
3. Concerns with Marker-Based System: The capture system utilizes three systems: pressure, vision, and an infrared marker-based system. However, the marker-based approach can interfere with the appearance in the vision data, making it unsuitable for monocular settings. From my perspective, in single-person scenarios without object interaction, there is typically less occlusion. Therefore, a multi-RGB capture setting (marker-less setting) should be sufficient for achieving motion capture quality.

**Suitability:**

3

---

### Meta-Review · Area_Chair_RXM5 · 2024-07-03

**Recommendation:** Accept (Poster)
**Confidence:** 5

**Metareview:**

This paper has a clear technical contribution in the collection of a dataset  integrating spatial-aware pressure information that is used to mitigate the depth ambiguity inherent in monocular 3D pose estimation solutions.The paper is very easy to follow and presents  solid experimental results.
All reviewers agreed on accepting the paper after the rebuttal. Therefore I reccommend acceptance.